# XOR-Based (*n*, *n*) Visual Cryptography Schemes for Grayscale or Color Images with Meaningful Shares

**Yu-Hong Chen and Justie Su-Tzu Juan \*** 

Department of Computer Science & Information Engineering, National Chi Nan University,
Puli, Nantou 54561, Taiwan
* Correspondence: jsjuan@ncnu.edu.tw

**Abstract:** XOR-based Visual Cryptography Scheme (XOR-based VCS) is a method of secret image sharing. The principle of XOR-based VCS is to encrypt a secret image into several encrypted images, called shares. No information about the secret can be obtained from any of the shares, and after applying the logical XOR operation to stack these shares, the original secret image can be recovered. In this paper, we present a new XOR-based VCS for grayscale or a color secret image. This scheme encrypts the secret grayscale (or color) image into *n* meaningful grayscale (or color) shares, which can import *n* difference cover images. After stacking *n* shares using the XOR operation, the original secret image can be completely restored. Both the theoretical proof and experimental results show that our method is accurate and efficient. To the best of our knowledge, ours is the only scheme that currently provides this functionality for grayscale and color secret images.

**Keywords:** secret sharing scheme; visual cryptography; XOR; meaningful shares; pixel expansion

## 1. Introduction

With the rapid development of network technology, information security is becoming more and more important. Visual Cryptography (VC) is one of the ways to strengthen information security. The basic concept of a Visual Cryptography Scheme (VCS) was proposed by Naor and Shamir [1] in 1995. The schematic of a VCS is shown in Figure 1. However, there are two disadvantages: pixel expansion and the codebook required. The random grid-based VCS [2] (RGVCS), proposed by Kafri and Keren in 1987, solves the issues of pixel expansion and the codebook required. Since then, the RGVCS has been extensively studied [3–9]. Note that aforementioned methods use Boolean OR operations to simulate the recovery operations in visual cryptography.

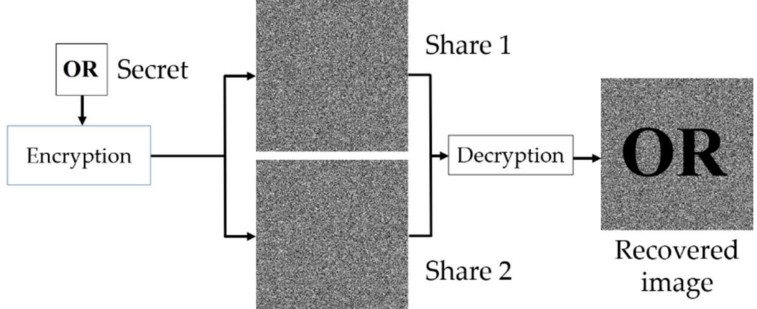

**Figure 1.** The flow chart of Naor and Shamir's VCS.

According to the principle, using the OR operation to recover shares allows at most one-half of the images to be completely restored. In order to increase the visual quality of the revealed image, the research into VCS began to focus on the XOR operation replacing

the OR operation as in the VCS; this is known as XOR-based VCS. Tuyls et al. show that a XOR-based VCS can be implemented using light polarization [10,11]. Thereafter, many related studies have been proposed [12–14].

In addition, all of the shares produced in [1,2] are meaningless. The researchers found that meaningless sharing schemes found it difficult to manage the shares and the encrypted trace on the shares was easily seen [6]. In order to solve the problems of meaningless shares and develop a XOR-based VCS, Wu and Sun, and Ou et al. proposed two XOR-based VCS methods to generate meaningful shares [15,16] separately. Their methods were to encrypt a secret into a cover image (the cover), and generate $n$ meaningful shares, which were similar to the cover (the same one). Finally, the secret could be restored by stacking $n$ shares using the XOR operation.

Lo and Juan [17] proposed three new methods in 2021 to optimize XOR-based VCS with meaningful shares, which was developed by Ou et al. in 2015 [16]. These methods encrypt the secret into $n$ covers (which can all be different or the same) and generate $n$ meaningful shares. According to the three methods, three shares of differing visual qualities are generated and the secret is revealed.

The previous methods are suitable for black-and-white images; therefore, in order to apply the previous methods to grayscale and color images, this paper extends the capability of the algorithms in [16,17]. Because black-and-white images are not popular nowadays, and people use grayscale and color images more often in modern society, we extended the ideas published in [16,17] to make the XOR-based VCS more practical.

The rest of the paper is organized as follows: Section 2 describes the studies related to our scheme, in which we explore the two methods related to the XOR-based VCS with meaningful shares. Section 3 described our proposed methods. We extended the capabilities of the algorithm proposed in [16,17] so that we were able to encrypt a secret grayscale (or color) image into $n$ meaningful grayscale (or color) shares that could be imported into $n$ different camouflage images. Section 4 describes the experimental results of our scheme. Section 5 describes the correctness and security analysis of our scheme. The conclusions are stated in Section 6.

## 2. XOR-Based VCS

In this section, we briefly introduce the XOR-based VCS proposed by Ou et al. [16] in 2015, which is related to the $(n, n)$ XOR-based VCS with meaningful shares, and the XOR-based VCS introduced by Lo and Juan [17] in 2021, which presents the extended capabilities of [16]. Before introducing these schemes, some VCS definitions are shown in Table 1.

**Table 1.** Notations used in this work.

| Notations | Description |
| --- | --- |
| $\oplus$ | Boolean XOR operation |
| $S$ | Secret image |
| $C_1, \ldots, C_n$ | Cover images |
| $R_1, \ldots, R_n$ | Shares generated by VCS |
| $(i, j)$ | The position of $i$ column and $j$ row |
| $S(i, j), C_x(i, j), R_x(i, j)$ | The value of $S(i, j)$, $C_x(i, j)$ and $R_x(i, j)$ for any $x \in \{1, 2, \ldots, n\}$ (for black-and-white image, or grayscale image) |
| $S^{R/G/B}(i, j), C_x^{R/G/B}(i, j), R_x^{R/G/B}(i, j)$ | The value of red/green/blue channel of $S(i, j)$, $C_x(i, j)$ and $R_x(i, j)$ for any $x \in \{1, 2, \ldots, n\}$ (for color image) |
| $R_{\{\oplus, 1, \ldots, n\}}$ | XOR-ed result by shares $R_1, \ldots, R_n$ |

### 2.1. The Ou et al.'s XOR-Based VCS

The schematic of Ou et al.'s XOR-based VCS [16] is shown in Figure 2. The principle described in [16] is to use Algorithm 1 to create two different arrays: $M_{n\text{-}odd}$ and $M_{n\text{-}even}$, which are used in Algorithm 2. In the Algorithm 3, first, the user sets a parameter $\beta$

($0 \leq \beta \leq 1$). When processing each pixel, the algorithm generates a random number $d$ ($0 \leq d \leq 1$). If $d \leq \beta$, it uses Algorithm 2 to encrypt this pixel. If $d > \beta$, it camouflages the pixels. Finally, XOR operation is used to stack all encrypted images, and the original secret image is completely restored.

---

**Algorithm 1.** Generate a $2^n \times n$ matrix for ($n$, $n$) XOR-based VCS [16]

---

**Input**: A parameter $n$.
**Output**: A $2^n \times n$ matrix $M_n$.

1.    **for** $i = 1; i <= 2^n; i = i + 1$ **do**

        $M_n(i, 1 : n) = \text{de2bi}(i - 1, n)$.

    **end for**

2.    Output the matrix $M_n$.

---

---

**Algorithm 2.** The basic algorithm for XOR-based VCS with meaningful shares [16]

---

**Input:** A binary secret image $S$ with $H \times W$ pixels, and a $2^n \times n$ matrix $M_n$.
**Output:** $n$ shares $R_1, \ldots, R_n$, each of which is $H \times W$ in size.

1.    Divide the matrix $M_n$ into two $2^{n-1} \times n$ sub-matrices $M_{n\text{-}odd}$ and $M_{n\text{-}even}$, where $M_{n\text{-}odd}$ includes the row vectors whose hamming weight is an odd number, while $M_{n\text{-}even}$ includes the row vectors whose hamming weight is an even number.

2.    **if** ($S(i, j) == 0$)

        Randomly select an integer value $r$ with equal probability on $[1, 2^{n-1}]$.

        $R_1(i, j) = M_{n\text{-}even}(r, 1), R_2(i, j) = M_{n\text{-}even}(r, 2), \ldots, R_n(i, j) = M_{n\text{-}even}(r, n)$.

    **else**

        Randomly select an integer value $r$ with equal probability on $[1, 2^{n-1}]$.

        $R_1(i, j) = M_{n\text{-}odd}(r, 1), R_2(i, j) = M_{n\text{-}odd}(r, 2), \ldots, R_n(i, j) = M_{n\text{-}odd}(r, n)$.

3.    Repeat Step 2 until all secret pixels have been processed, and output $n$ shares $R_1, \ldots, R_n$, each of which has the same image size as the secret image.

---

---

**Algorithm 3.** ($n$, $n$) XOR-based VC scheme with meaningful shares [16]

---

**Input:** A binary secret image $S$ and a cover image $C$, both with $H \times W$ pixels, and a parameter $\beta$.
**Output:** $n$ meaningful shares $R_1, \ldots, R_n$, each of which is $H \times W$ in size.

1.    Generate a random floating point $d$ ($0 \leq d \leq 1$).

2.    **if** ($d < \beta$)

        Generate $n$ pixels $t_1, \ldots, t_n$ by **Algorithm 2** for secret pixel $S(i, j)$.

        $R_1(i, j) = t_1, R_2(i, j) = t_2, \ldots, R_n(i, j) = t_n$.

3.    **else**

        **if**($n \times C(i, j)$ is an even number)

            $R_1(i, j) = C(i, j), R_2(i, j) = C(i, j), \ldots, R_n(i, j) = C(i, j)$.

        **else**

            Randomly select a number $f$ from $\{1, \ldots, n\}$.

            $R_1(i, j) = C(i, j), \ldots, R_f(i, j) = 1 - C(i, j), \ldots, R_n(i, j) = C(i, j)$.

4.    Repeat Steps 1-3 until all secret pixels have been processed, and output $n$ shares $R_1, \ldots, R_n$, each of which has the same image size as the secret image.

---

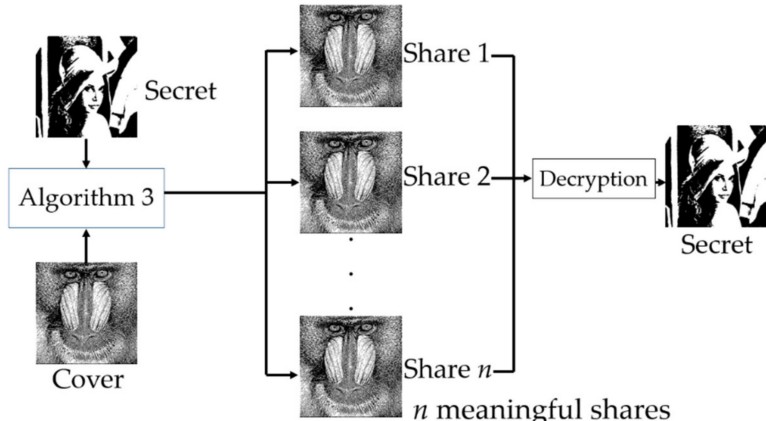

**Figure 2.** The basic schematic of Ou et al.'s XOR-based VCS.

## 2.2. The Lo and Juan's XOR-Based VCS

Lo and Juan [17] proposed three new methods that inherit the advantages of the method proposed in [16]. The schematic from [17] is shown in Figure 3. The difference between [16] and [17] is that Lo and Juan's methods can input $n$ covers (the covers can all be different or the same), while [16] can only input one cover. Among their three methods, Algorithm 4 [new method 2] can completely reconstruct the original secret. To be able to import multiple cover images, some steps in Algorithm 3 are changed, as shown below.

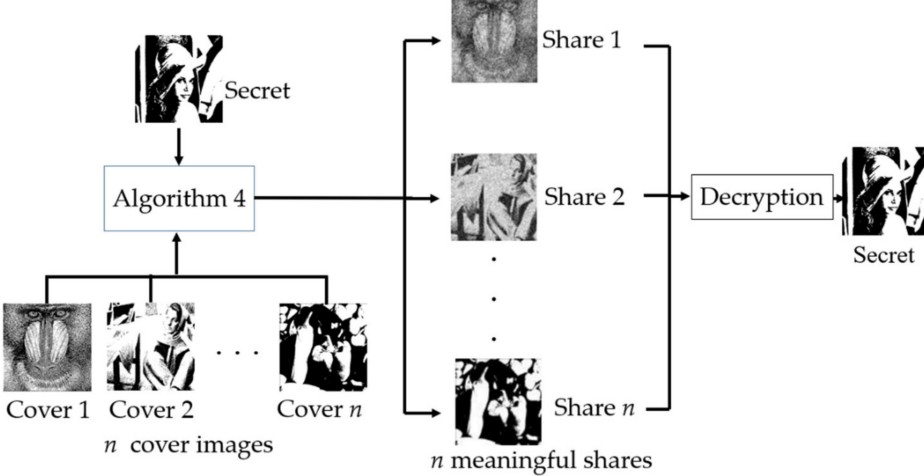

**Figure 3.** The basic schematic of Lo and Juan's XOR-based VCS.

---

**Algorithm 4 [New method 2].** $(n, n)$ XOR-based VC scheme with meaningful shares [17]

---

**Input**: A binary secret image $S$ and $n$ cover image $C_1, \ldots, C_n$, all of them with $H \times W$ pixels, and a parameter $\beta$.

**Output**: $n$ meaningful shares $R_1, \ldots, R_n$, each of which is $H \times W$ in size.

1.  Generate a random floating point $d$ ($0 \leq d \leq 1$).
2.  **if** ($d < \beta$)

    Generate $n$ pixels $t_1, \ldots, t_n$ by **Algorithm 2** for secret pixel $S(i, j)$.

    $R_1(i, j) = t_1, R_2(i, j) = t_2, \ldots, R_n(i, j) = t_n$
3.  **else**

    **if** ($C_1(i, j) \oplus C_2(i, j) \oplus \ldots \oplus C_n(i, j) = S(i, j)$)

    $R_1(i, j) = C_1(i, j), R_2(i, j) = C_2(i, j), \ldots, R_n(i, j) = C_n(i, j)$

    **else**

    Randomly select a number $f$ from $\{1, \ldots, n\}$.

    $R_1(i, j) = C_1(i, j), \ldots, R_f(i, j) = 1 - C_f(i, j), \ldots, R_n(i, j) = C_n(i, j)$
4.  Repeat Steps 1-3 until all secret pixels have been processed, and output $n$ shares $R_1, \ldots, R_n$, each of which has the same image size as the secret image.

---

## 3. Main Results

The advanced XOR-based $(n, n)$ VCS with meaningful shares is presented in this section. In Lo and Juan's study [17], the user sets the parameter $\beta$ ($0 \leq \beta \leq 1$) first. This denotes what percentage of the secret image is encrypted in order to keep the secret (otherwise those pixels will be camouflaged). If it is decided that one pixel is encrypted to keep the secret, the encrypted pixels for the secret are calculated and put in the shares. Therefore, the higher the $\beta$, the more meaningless the shares are, and the contrast of the revealed secret will be higher. On the other hand, if it is decided that one pixel is encrypted for camouflage, the encrypted pixels for the cover and secret images are also calculated and put in the shares. Most pixel of those shares look the same as the covers. If the number of camouflage pixels is more than encrypted pixels, the share is similar to the cover, and the contrast of the reveal secret will be lower. In [17], both methods dealing with binary images. In order to keep the advantages and enhance the ability of both methods, our aim was to camouflage every pixel and completely restore the secret image after stacking using the XOR operation for grayscale and color images. As a result, in order to achieve our purpose, we did not use Algorithm 1 or 2 in [16], but modified Algorithm 4 in [17], which was mentioned in the previous section.

### 3.1. (n, n) XOR-Based VCS with Grayscale Secret and Covers

For reasons of camouflage, we set the share as the cover image; but in order to completely restore the original secret image after applying the XOR operation on all $n$ shares, we need to change some pixels (of some shares), which makes those shares different to the cover image and negates the purpose of the camouflage. Therefore, in order to minimize this, we want to change only one share pixel, and choose the one that changes the least.

The schematic of Algorithm 5 is shown in Figure 4, and the flow chart of Algorithm 5 is shown in Figure 5. In Algorithm 5, for each pixel of the secret image, we calculate the value $P_x(i, j)$ for each cover first; then, we choose the minimum one to be $P_{min}(i, j)$ from all $x$ in $\{1, 2, \ldots, n\}$. If more than two shares obtain the same minimum value, we randomly choose a share with equal probability to be *min*. Then, the *min*-th share pixel is set to the value of $P_x(i, j)$. Other $n - 1$ shares are set to be $C(i, j)$. Therefore, after doing the XOR operation with other $n - 1$ shares, $S(i, j)$ is obtained.

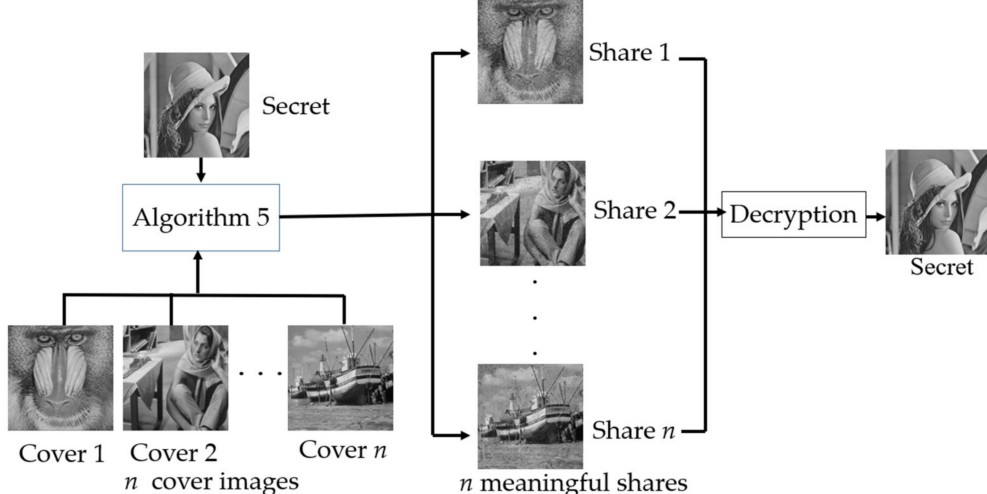

**Figure 4.** The basic schematic of ($n$, $n$) XOR-based VCS with grayscale secret and covers.

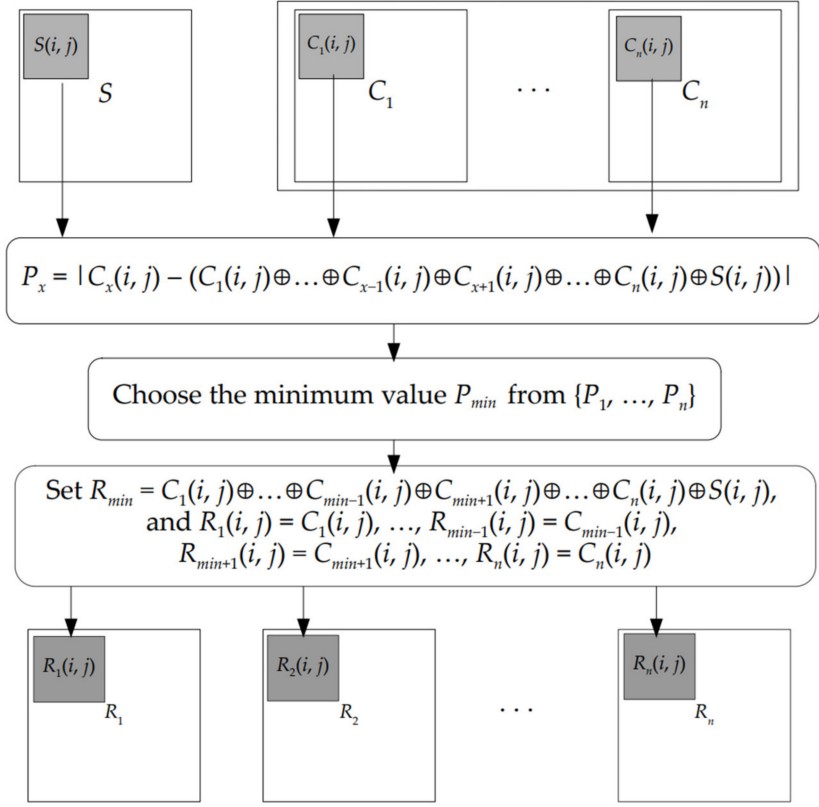

**Figure 5.** The flow chart of Algorithm 5.

*3.2. ($n$, $n$) XOR-Based VCS with Color Secret and Covers*

Similarly, in Algorithm 6, for the red, blue, and green channels, we calculate the value $P_x^R(i, j)$, $P_x^G(i, j)$, $P_x^B(i, j)$ for each pixel of each cover, and choose the minimum $P_{\min}^R$, $P_{\min}^G$, $P_{\min}^B$ from all $x$ in $\{1, 2, \ldots, n\}$ separately. In each channel, if more than two values are identical and are also the minimum value, we randomly choose a share with an equal probability to change the pixel. Then, for each channel $A$ ($A = R, G$ or $B$), the *min*-th share pixel is set to the value produced by XORing $S^A(i, j)$ with the other $n - 1$ pixels. Other $n - 1$ shares $R_x^R(i, j)$, $R_x^G(i, j)$, $R_x^B(i, j)$ are set to be $C_x^R(i, j)$, $C_x^G(i, j)$, $C_x^B(i, j)$, respectively. The schematic and the flow chart of Algorithm 6 are shown in Figures 6 and 7, respectively.

---

**Algorithm 5.** $(n, n)$ XOR-based VC scheme with grayscale meaningful shares.

---

**Input**: A secret grayscale image $S$ and $n$ cover grayscale image $C_1, \ldots, C_n$, all of them with $H \times W$ pixels.

**Output**: $n$ gray meaningful shares $R_1, \ldots, R_n$, each of which is $H \times W$ in size.

1. Choose one position $(i, j)$ from $H \times W$.
2. For each covers $C_1(i, j), \ldots, C_n(i, j)$, calculate the value $P_x(i, j) = |\,C_x(i, j) - (C_1(i, j) \oplus \ldots \oplus C_{x-1}(i, j) \oplus C_{x+1}(i, j) \oplus \ldots \oplus C_n(i, j) \oplus S(i, j))\,|$.
3. Choose $min$ from $\{1, 2, \ldots, n\}$ such that $P_{min}(i, j) = \min\{P_1(i, j), P_2(i, j), \ldots, P_n(i, j)\}$.
4. $R_{min}(i, j)$ is set to $R_{min}(i, j) = C_1(i, j) \oplus \ldots \oplus C_{min-1}(i, j) \oplus C_{min+1}(i, j) \oplus \ldots \oplus C_n(i, j) \oplus S(i, j)$. Other $n - 1$ share pixels are set to $C(i, j)$, such that $R_1(i, j) = C_1(i, j), \ldots, R_{min-1}(i, j) = C_{min-1}(i, j)$, $R_{min+1}(i, j) = C_{min+1}(i, j), \ldots, R_n(i, j) = C_n(i, j)$.
5. Repeat Steps 1-4 until all secret pixels have been processed, and output $n$ shares $R_1, \ldots, R_n$, each of which has the same image size as the secret image.

---

**Algorithm 6.** $(n, n)$ XOR-based VC scheme with color meaningful shares

---

**Input**: A secret color image $S$ and $n$ cover color image $C_1, \ldots, C_n$, all of them with $H \times W$ pixels.

**Output**: $n$ color meaningful shares $R_1, \ldots, R_n$, each of which is $H \times W$ in size.

1. Choose one position $(i, j)$ from $H \times W$.
2. For each channel $A \in \{R, G, B\}$ do Step 3-5.
3. For each $x \in \{1, 2, \ldots, n\}$, calculate the value
   $P_x^A(i, j) = |\,C_x^A(i, j) - (C_1^A(i, j) \oplus \ldots \oplus C_{x-1}^A(i, j) \oplus C_{x+1}^A(i, j) \oplus \ldots \oplus C_n^A(i, j) \oplus S^A(i, j))\,|$.
4. Choose $min$ from $\{1, 2, \ldots, n\}$ such that $P_{min}^A = \min\{P_1^A(i, j), P_2^A(i, j), \ldots, P_n^A(i, j)\}$.
5. $R_{min}^A(i, j)$ is set to be $R_{min}^A(i, j) = C_1^A(i, j) \oplus \ldots \oplus C_{min-1}^A(i, j) \oplus C_{min+1}^A(i, j) \oplus \ldots \oplus C_n^A(i, j) \oplus S^A(i, j)$. Finally, other $n - 1$ share pixels are set to be $C^A(i, j)$, such that $R_1^A(i, j) = C_1^A(i, j), \ldots,$ $R_{min-1}^A(i, j) = C_{min-1}^A(i, j), R_{min+1}^A(i, j) = C_{min+1}^A(i, j), \ldots, R_n^A(i, j) = C_n^A(i, j)$.
6. Repeat Steps 1-5 until all secret pixels have been processed, and output $n$ shares $R_1, \ldots, R_n$, each of which has the same image size as the secret image.

---

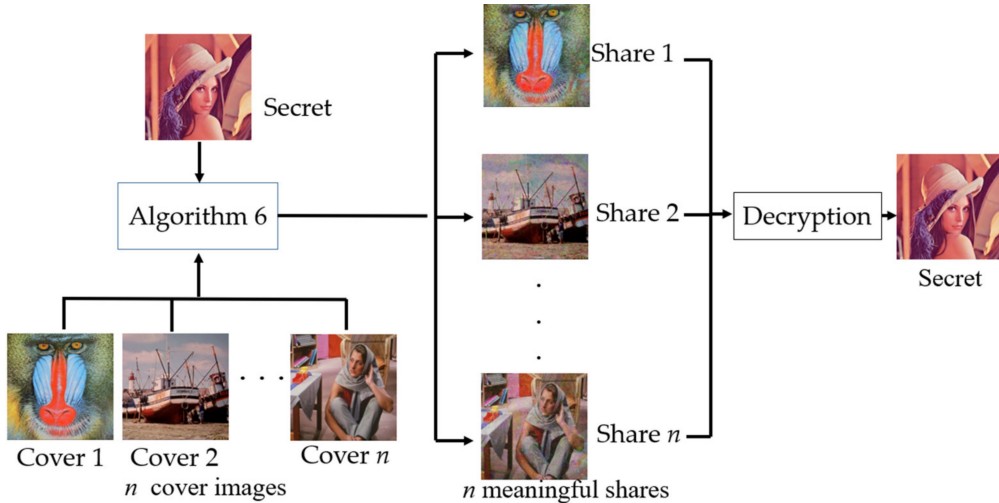

**Figure 6.** The basic schematic of Algorithm 6.

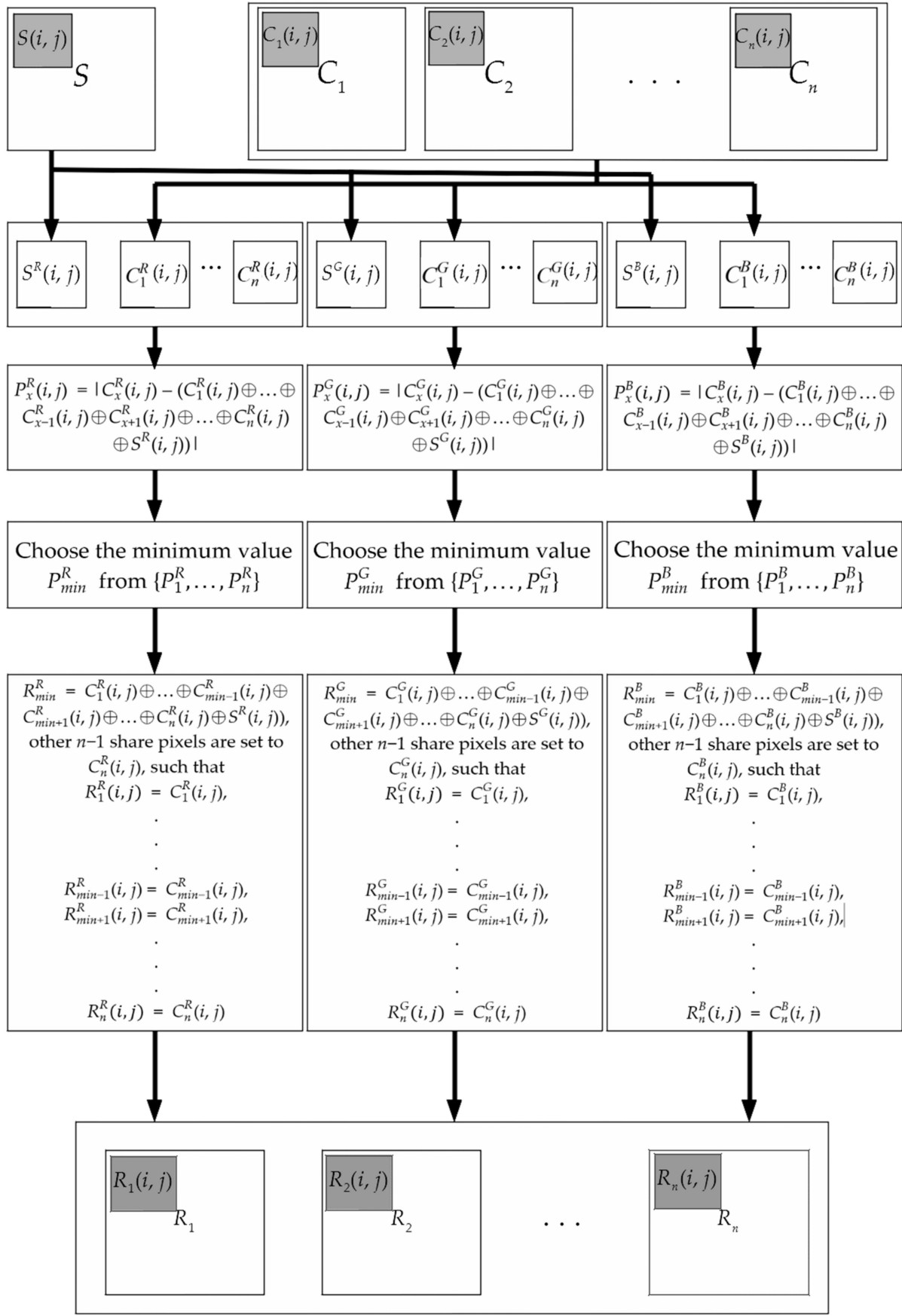

**Figure 7.** The flow chart of Algorithm 6.

## 4. Experimental Results

In this section, we present the results of various experiments related to the proposed algorithms. We illustrate the experimental results for encrypting a secret image into *n* cover images and analyze the PSNR between covers and shares.

### 4.1. Objective Evaluation Parameters

This paper uses two objective evaluation parameters: Mean Square Error (MSE) and Peak Signal to Noise Ratio (PSNR). The MSE is an estimator, which means the average squared difference between the experiment values and the original value. The PSNR is the image quality assessment metric. *H* and *W* denote the height and weight of images. The secret image, the restored image, cover images, and share images are the same size. The MSE and PSNR are defined in the following. Because the MSE is a risk function, corresponding to the expected value of the squared error loss, it is always strictly positive and unlimited. The PSNR ranges from 0 to 50 and is expected to be as high as possible. Normally, if the PSNR is greater than 30, the encrypted traces on the shares cannot be detected with the human eyes. For color images with RGB values per pixel, the MSE is the sum over all squared R, G, and B value differences and the PSNR is the MSE divided by the image size and by three. For two images *A*, *B* (a secret image and a restored image; or the cover image and share), the PSNR is defined as

$$\text{MSE} = \frac{\sum_{i=1}^{H} \sum_{j=1}^{W} (A(i, j) - B(i, j))^2}{H \times W} \tag{1}$$

$$\text{PSNR} = 10 \times \log\left(\frac{255^2}{MSE}\right) \text{(dB)} \tag{2}$$

### 4.2. The Experimental Results of the Proposed VCS with Grayscale Images

The experimental results from Algorithm 5 for the case of (5, 5) are shown in Figure 8. All of the grayscale images are 512 × 512 pixels in size. Figure 8a shows the secret image. Figure 8b–f show the cover images. Figure 8h–l show the share images generated by Algorithm 5. Figure 8g shows the revealed image via XORing all shares.

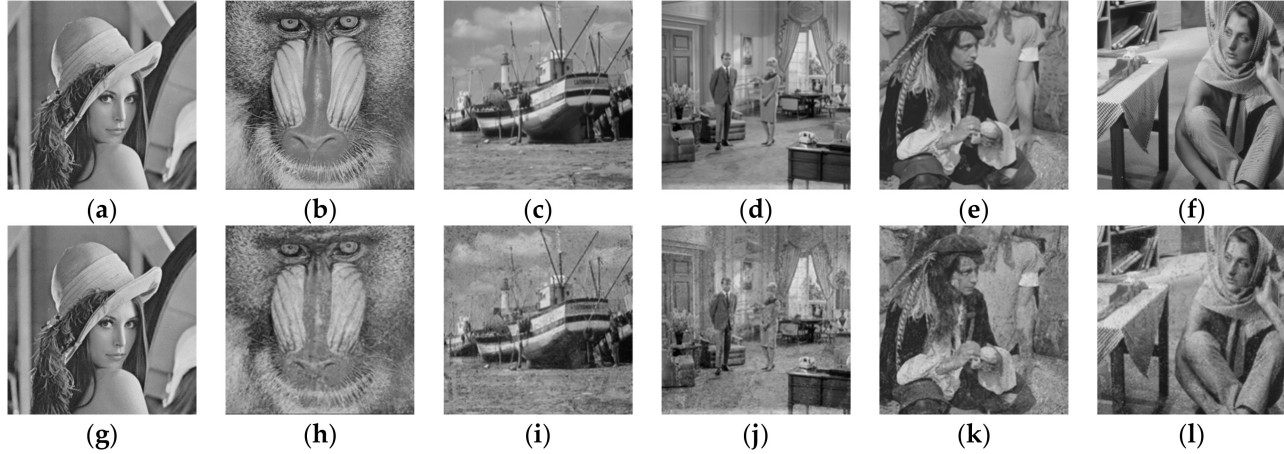

**Figure 8.** Experimental results from Algorithm 5 where *n* = 5 and all image sizes are 512 × 512: (**a**) secret image (**b**) $C_1$; (**c**) $C_2$; (**d**) $C_3$; (**e**) $C_4$; (**f**) $C_5$; (**g**) $R_1 \oplus R_2 \oplus R_3 \oplus R_4 \oplus R_5$; (**h**) $R_1$; (**i**) $R_2$; (**j**) $R_3$; (**k**) $R_4$; (**l**) $R_5$.

Table 2 illustrates the PSNR of the images in Figure 8. We found that all of the PSNR values were between 19 to 20 and the original secrets were perfectly reconstructed. All of the PSNR values of our experimental results were less than 30, and we found a little

encrypted trace on the share images. However, we could still easily see that the shares were very similar to the camouflage images.

**Table 2.** The PSNR of the reveal image and shares generated by Algorithm 5 for the (5, 5) case.

| Images | PSNR |
|--------|------|
| R1 | 19.7081 |
| R2 | 19.4408 |
| R3 | 19.6550 |
| R4 | 19.5888 |
| R5 | 19.7651 |

To better understand the experimental results from Algorithm 5, we chose *n* images from 15 images as covers: *n* represents the number of shares, which were 3, 4, 5, 6, and 7. The 15 images are shown in Figure 9a–o (from https://pixabay.com, accessed date: 5 July 2022). After all possible experiments were performed, the average PSNR value was calculated. Under this assumption, we performed $C(15, n)$ times for each case to calculate average PSNR. The experimental results are shown in Table 3. From Table 3, we can obviously see that as *n* increases, the average visual quality of shares becomes better (increases).

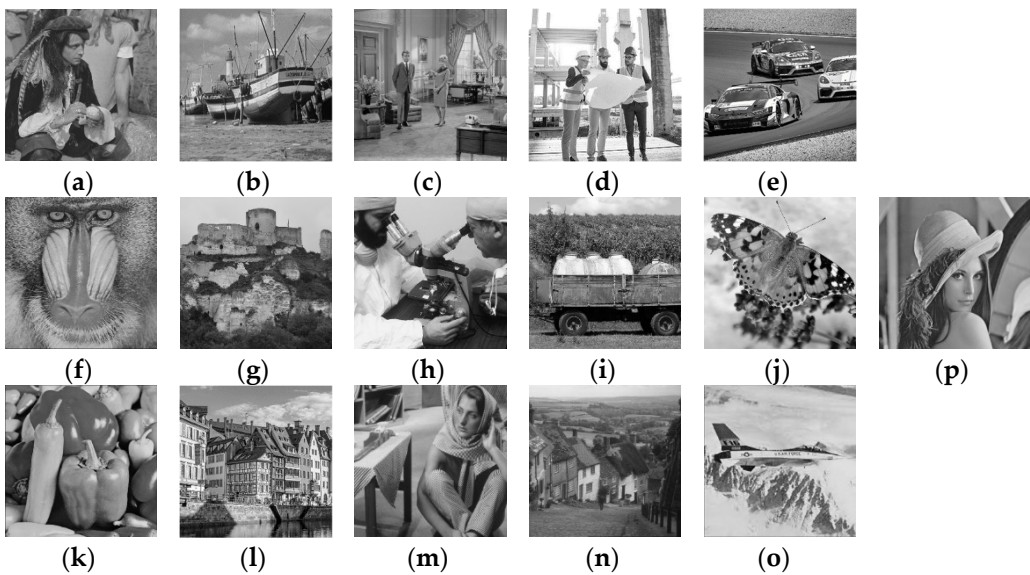

**Figure 9.** The secret and 15 different covers: (**a**) man; (**b**) boat; (**c**) couple; (**d**) architect; (**e**) cars; (**f**) baboon; (**g**) castle; (**h**) scientists; (**i**) agriculture; (**j**) butterfly; (**k**) pepper; (**l**) houses; (**m**) Barbara; (**n**) hill; (**o**) jet; (**p**) secret.

**Table 3.** The average PSNR of shares in (3, 3), (4, 4), (5, 5), (6, 6), (7, 7) cases, with one secret and 15 different grayscale covers, as shown in Figure 9.

| Cases (*n*, *n*) | $C(15, n)$ | Average PSNR |
|------------------|------------|--------------|
| (3, 3) | 455 | 16.4814 |
| (4, 4) | 1365 | 18.1383 |
| (5, 5) | 3003 | 19.4192 |
| (6, 6) | 5005 | 20.3865 |
| (7, 7) | 6435 | 21.1766 |

### 4.3. The Experimental Results of the Proposed VCS with Color Images

Similarly, the experimental results from Algorithm 6 for the (5, 5) case are shown in Figure 10. All of the color images are 512 × 512 pixels in size. Figure 10a shows the secret

image. Figure 10b–f show the cover images. Figure 10h–l show the share images generated by Algorithm 6. Figure 10g shows the revealed image via XORing all shares. Although there are some traces of encryption in the shares, we were still clearly able to analyze the meaning on the shares. Table 4 illustrates the PSNR of the images in Figure 10, all with a PSNR of between 19 to 20. We also calculated $C(15, n)$ times to obtain the average PSNR in different cases. The 15 images are shown in Figure 11 (from https://pixabay.com, accessed date: 5 July 2022).

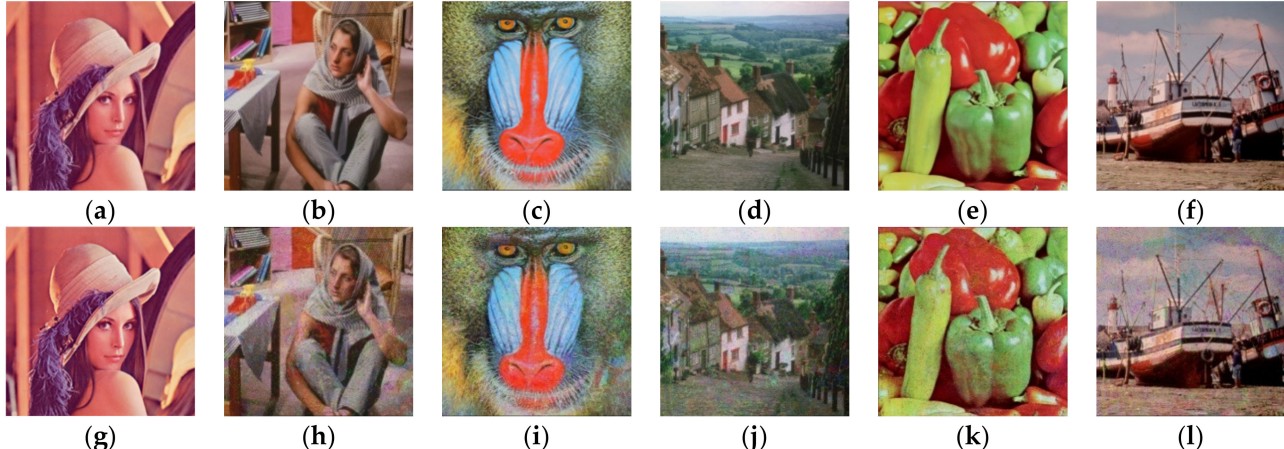

$$\begin{array}{cccccc} \text{(a)} & \text{(b)} & \text{(c)} & \text{(d)} & \text{(e)} & \text{(f)} \\ \text{(g)} & \text{(h)} & \text{(i)} & \text{(j)} & \text{(k)} & \text{(l)} \end{array}$$

**Figure 10.** Experimental results from Algorithm 6 where $n = 5$ and all size of images are $512 \times 512$: (**a**) secret image (**b**) $C_1$; (**c**) $C_2$; (**d**) $C_3$; (**e**) $C_4$; (**f**) $C_5$; (**g**) $R_1 \oplus R_2 \oplus R_3 \oplus R_4 \oplus R_5$; (**h**) $R_1$; (**i**) $R_2$; (**j**) $R_3$; (**k**) $R_4$; (**l**) $R_5$.

**Table 4.** The PSNR of the reveal image and shares generated by Algorithm 6 for the (5, 5) case.

| Images | PSNR |
|:---:|:---:|
| *R1* | 19.5757 |
| *R2* | 19.1196 |
| *R3* | 19.2015 |
| *R4* | 19.4861 |
| *R5* | 19.1398 |

The experimental results are shown in Table 5. From Table 5, we can obviously see that as $n$ increases, the average visual quality of shares becomes better. It is easy to see that the average PSNR in Table 5 is similar to the value in Table 3 for each case.

**Table 5.** The average PSNR of shares in the (3, 3), (4, 4), (5, 5), (6, 6), (7, 7) cases, with one secret and 15 different color covers, as shown in Figure 11.

| Cases (*n*, *n*) | *C*(15, *n*) | Average PSNR |
|:---:|:---:|:---:|
| (3, 3) | 455 | 16.2671 |
| (4, 4) | 1365 | 18.0392 |
| (5, 5) | 3003 | 19.2421 |
| (6, 6) | 5005 | 20.3442 |
| (7, 7) | 6435 | 21.4743 |

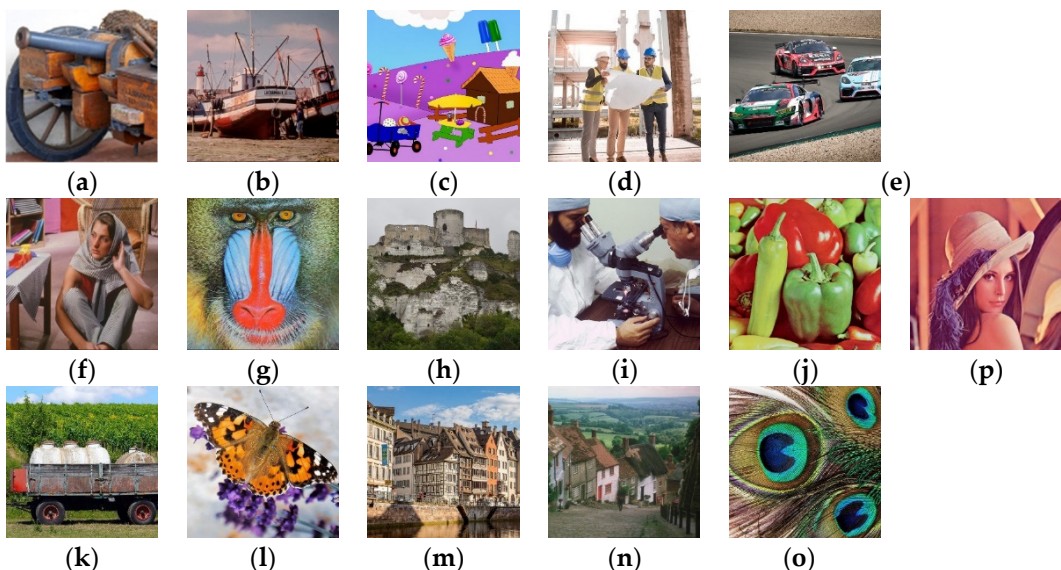

**Figure 11.** The secret and 15 different covers: (**a**) cannon; (**b**) boat; (**c**) cartoon; (**d**) architect; (**e**) cars; (**f**) Barbara; (**g**) baboon; (**h**) castle; (**i**) scientists; (**j**) pepper; (**k**) agriculture; (**l**) butterfly; (**m**) houses; (**n**) hill; (**o**) feather; (**p**) secret.

## 5. Discussion

### 5.1. Correctness Analysis

According to the two proposed algorithms, when one collects all $n$ shares and applying XOR operation on all shares, the secret image can be reconstructed completely. That is because we set $R_{min}(i, j) = C_1(i, j) \oplus \ldots \oplus C_{min-1}(i, j) \oplus C_{min+1}(i, j) \oplus \ldots \oplus C_n(i, j) \oplus S(i, j)$, and other $n - 1$ share pixels are set to $C(i, j)$ in Algorithm 5; $R_{min}^A(i, j)$ is set to be $R_{min}^A(i, j) = C_1^A(i, j) \oplus \ldots \oplus C_{min-1}^A(i, j) \oplus C_{min+1}^A(i, j) \oplus \ldots \oplus C_n^A(i, j) \oplus S^A(i, j)$, and other $n - 1$ share pixels are set to be $C^A(i, j)$ for any $A \in \{R, G, B\}$ in Algorithm 6. Therefore, $R_1(i, j) \oplus R_2(i, j) \oplus \ldots \oplus R_n(i, j) = C_1(i, j) \oplus \ldots \oplus C_{min-1}(i, j) \oplus (C_1(i, j) \oplus \ldots \oplus C_{min-1}(i, j) \oplus C_{min+1}(i, j) \oplus \ldots \oplus C_n(i, j) \oplus S(i, j)) \oplus C_{min+1}(i, j) \oplus \ldots \oplus C_n(i, j) = S(i, j)$ in Algorithm 5. In the same way, $R_1^A(i, j) \oplus R_2^A(i, j) \oplus \ldots \oplus C_n^A(i, j) = S^A(i, j)$ for any $A \in \{R, G, B\}$ in Algorithm 6. This means that the recovered image is completely equal to the secret image in both algorithms, and so the proposed schemes are correct.

### 5.2. Security Analysis

In the proposed scheme, we need to stack all shares to fully recover the original secret image; otherwise, the secret image cannot be seen. For each pixel, only one share is encrypted, the others are the same as the cover. When we only stack $k$ ($k < n$) shares, if the encrypted shares are not collected, it is the same as what we do for the XOR operation for $k$ random values, so the original secret image cannot be reconstructed. Similarly, if we have the encrypted share but lack any other share on these $k$ shares, a random value is lost, so we cannot fully recover the original secret image. If we try to guess the value of an encrypted share or missing share, it is equivalent to guessing a random value in the range of 0 to 255. As a result, we can ensure that only those who have all the shares can reconstruct the secret. This means that the proposed schemes are secure. Moreover, binary (black-and-white) images are not common nowadays. Therefore, if we have binary images on our cellphone or computer, it may attract malicious attacks. Hence, the proposed schemes are more practical and safer than existing schemes

In the proposed scheme, the pixels used in all but one shares are equal to the pixels of the cover images, and the pixel of the last share is calculated solely from the pixels of the secret image and all other shares. Therefore, if one cannot obtain all the shares, the secret image cannot be restored. To demonstrate that our scheme is secure, we stacked the

partial, incomplete *k* shares in Figures 8 and 10 for observation (*k* < *n*). Figure 12a shows that we stacked two shares: Figure 8h,i. Figure 12b shows that we stacked another two shares: Figure 8k,l. Figure 12c shows that we stacked three shares: Figure 8h–j. Figure 12d shows that we stack another three shares: Figure 8h,i,k. Figure 12e shows that we stacked four shares: Figure 8h–j,l. Figure 12f shows that we stacked another four shares: Figure 8i–l. Figure 12g shows that we stacked two shares: Figure 10i,k. Figure 12h shows that we stacked another two shares: Figure 10h,j. Figure 12i shows that we stacked three shares: Figure 10h–j. Figure 12j shows that we stacked another three shares: Figure 10j–l. Figure 12k shows that we stack four shares: Figure 10h–j,l. Figure 12l shows that we stack another four shares: Figure 10h,j–l.

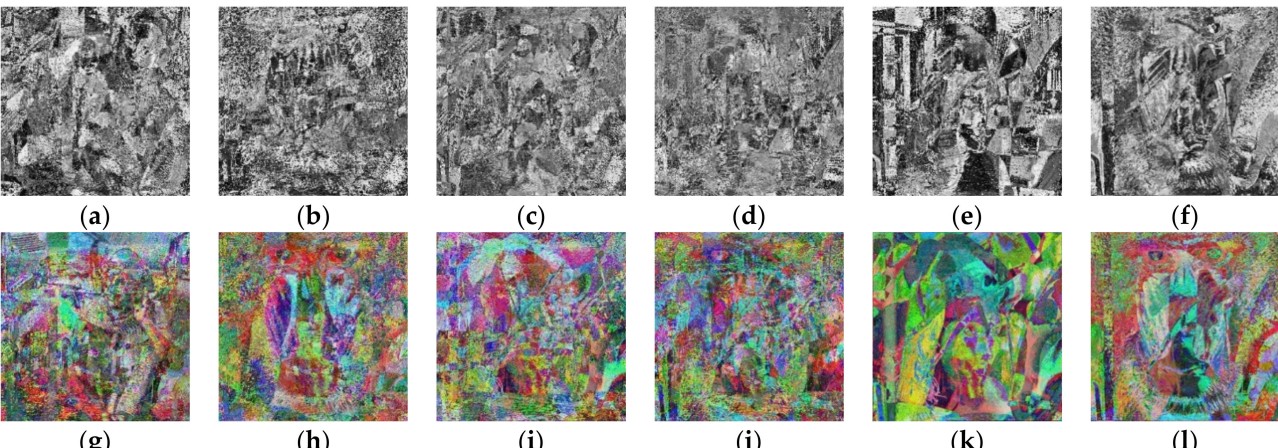

**Figure 12.** Stacking *k* images in Figures 8 and 10, where *k* = 2, 3 ,4: (**a**) Figure 8h⊕Figure 8i; (**b**) Figure 8k⊕Figure 8l; (**c**) Figure 8h⊕Figure 8i⊕Figure 8j; (**d**) Figure 8h⊕Figure 8k⊕Figure 8l; (**e**) Figure 8h⊕Figure 8i⊕Figure 8j⊕Figure 8l; (**f**) Figure 8i⊕Figure 8j⊕Figure 8k⊕Figure 8l; (**g**) Figure 10i⊕Figure 10k; (**h**) Figure 10h⊕Figure 10j; (**i**) Figure 10h⊕Figure 10i⊕Figure 10j; (**j**) Figure 10j⊕Figure 10k⊕Figure 10l; (**k**) Figure 10h⊕Figure 10i⊕Figure 10j⊕Figure 10l; (**l**) Figure 10h⊕Figure 10j⊕Figure 10k⊕Figure 10l.

From Figure 12, we can see that if we only stacked *k* (*k* = 2, 3, 4) shares, it generated a meaningless image. Because we did not stack all of the shares, these stacked images lost their meaning. As a result, we were able to ensure that our scheme was secure since the secret image could not be reconstructed until all shares were collected.

*5.3. Theoretical and Experimental PSNR*

To be able to theoretically evaluate the PSNR of the general case, some definitions were introduced to substantiate the analysis of the proposed methods. In Step 2 of Algorithm 5 and Step 3 of Algorithm 6, we calculated the difference between the pixel of the *x*-th camouflage image $C_x(i, j)$ and its "target" ($C_1(i, j)⊕ \ldots ⊕C_{x-1}(i, j)⊕C_{x+1}(i, j)⊕ \ldots ⊕C_n(i, j)⊕S(i, j)$). Because we had no idea about the difference, it was necessary to calculate the expectation value of this difference, and use it to obtain a theoretical low bound of the PSNR of each share in the proposed scheme. For any possible secret and camouflage image, each pixel of the secret and target can be seen as two random values. The expectation value *E* for the difference between them is defined as

$$E = \left( \sum_{i=0}^{255} \left( \sum_{j=0}^{255} |i - j| \right) \right) / 256^2 \tag{3}$$

After calculation, we obtained *E* to be about 85.332031. If the target pixels were all encrypted in the same single share, we were able to estimate the MSE between this cover image and its share as follows:

$$S\_\text{MSE} = \frac{\sum_{i=1}^{H} \sum_{j=1}^{W} E^2}{H \times W} = 7281.5534. \tag{4}$$

However, our proposed method selects the minimum among the $n$ shares as the target pixel for each pixel, so the actual MSE of the proposed scheme is less than or equal to the theoretical MSE ($T\_\text{MSE}$), which considers the target pixel to be uniformly selected from all $n$ shares. Therefore, only $1/n$ pixels of any one share are selected as the target pixels, and $T\_\text{MSE}$ is defined as follows:

$$T\_\text{MSE} \left( \frac{\sum_{i=1}^{H} \sum_{j=1}^{W} E^2}{H \times W} \right) / n \tag{5}$$

According to the theoretical MSE, one can obtain the theoretical PSNR ($T\_\text{PSNR}$) as follows:

$$T\_\text{PSNR} = 10 \times \log\left( \frac{255^2}{T\_\text{MSE}} \right) \text{(dB)} \tag{6}$$

Table 6 shows the $T\_\text{MSE}$, $T\_\text{PSNR}$ and experimental PSNR, which is equal to Table 3, for $n = 3$ to 7 of Algorithm 5. Similarly, Table 7 shows the $T\_\text{MSE}$, $T\_\text{PSNR}$, and experimental PSNR, which are quoted from Table 7, for $n = 3$ to 7 of Algorithm 6. From Tables 6 and 7, we can observe that the experimental PSNR in each case is larger than the theoretical PSNR.

**Table 6.** The theoretical MSE, PSNR, and experimental PSNR for $n = 3$ to 7 of Algorithm 5.

| Cases | Theoretical MSE | Theoretical PSNR | Experimental PSNR |
|-------|-----------------|------------------|-------------------|
| (3, 3) | 2427.1851 | 14.2797 | 16.4814 |
| (4, 4) | 1820.3888 | 15.5291 | 18.1383 |
| (5, 5) | 1456.3110 | 16.4982 | 19.4192 |
| (6, 6) | 1213.5925 | 17.2900 | 20.3865 |
| (7, 7) | 1040.2221 | 17.7959 | 21.1766 |

**Table 7.** The theoretical MSE, PSNR, and experimental PSNR for $n = 3$ to 7 of Algorithm 6.

| Cases | Theoretical MSE | Theoretical PSNR | Experimental PSNR |
|-------|-----------------|------------------|-------------------|
| (3, 3) | 2427.1851 | 14.2797 | 16.2671 |
| (4, 4) | 1820.3888 | 15.5291 | 18.0392 |
| (5, 5) | 1456.3110 | 16.4982 | 19.2421 |
| (6, 6) | 1213.5925 | 17.2900 | 20.3442 |
| (7, 7) | 1040.2221 | 17.7959 | 21.4743 |

*5.4. Comparison with State-of-the-Art Approaches*

In this section, we compare the capabilities of the proposed schemes with those of state-of-the-art approaches. We list seven parts of each scheme to assist in the evaluation. The criteria are illustrated as follows:

1. Type of images: In other schemes, the experimental images are all black-and-white or halftone images. There is only 1 bit used per pixel in each image. In our scheme, we use grayscale and color images, and the bits per pixel are 8 and 24, respectively.
2. Pixel expansion: This refers to whether the scheme suffers from the pixel expansion problem. Pixel expansion increases the transmission time and decreases bandwidth utilization. There is no pixel expansion in our scheme.
3. Number of cover image: This refers to how many covers can be imported. The higher the cover number, the more situations can be used.
4. The number of share images: In our scheme, the amount of covers we import denotes how many shares we can generate. From Table 8, we can obviously see that our scheme is the same as the others.

5. Meaningful share: This refers to whether the scheme can generate a meaningful share. Meaningful shares are more desirable than meaningless shares, because they look more natural and other people will not be attracted by them.
6. Decryption operation: There are two operations for any VCS, OR, and XOR operation. The secret image can be completely reconstructed using the XOR operation, while at most one-half of the image can be reconstructed by the OR operation.
7. Contrast: As defined in [3,16–18], this decides how human eyes recognize the reconstructed image. It ranges from 0 to 1. The higher the value, the more easily the restored image can be identified.

**Table 8.** Comparisons of capabilities of our scheme, Noar and Shamir's scheme [1], Ou et al.'s scheme [16], Singh et al.'s scheme [18], and Lo and Juan's scheme [17].

| Schemes | Type of Image | Pixel Expansion | Number of Cover Image | Number of Share Image | Meaningful Share | Decryption Operation | Contrast |
|---|---|---|---|---|---|---|---|
| [1] | Black and white | Yes | 0 | $n$ | No | OR operation | 1/2 |
| [16] | Black and white | No | 1 | $n$ | Yes | XOR operation | 1 |
| [18] | Black and white | No | $n/2$ | $n$ | Yes | XOR operation | 1 |
| [17] | Black and white | No | $n$ | $n$ | Yes | XOR operation | 1 |
| Ours | Grayscale and color | No | $n$ | $n$ | Yes | XOR operation | 1 |

Table 8 shows a comparisons between the capabilities of our schemes and those of [1,16–18]. Compared to our schemes, that in [1] has the problem of pixel expansion. Moreover, the scheme in [1] cannot generate meaningful shares, and the contrast of the restored secret is only one-half. It can be easily observed that the number of cover images of the schemes in [1,16,18] are 0, 1, and $n/2$, respectively. The main difference between our schemes and that in [17] is that our schemes work on grayscale and color images. We can apply our schemes to a wider variety of situations.

## 6. Conclusions

Without a doubt, the visual cryptography scheme is one of the safest ways by which to protect confidential images and transfer them via mobile phones, computers, or over the Internet. In this paper, we propose advanced XOR-based $(n, n)$ visual cryptographic schemes with meaningful sharing. Unlike other VCS studies that focus on binary images, our study proposes two algorithms that can handle grayscale and color images. Our scheme can encrypt a secret grayscale (or color) image into $n$ meaningful grayscale (or color) shares that can be imported into $n$ different camouflage images. The theoretical analysis and experimental results showed that both of our schemes can completely recover the original secret image using the XOR operation and that they are secure. Our method can be applied to more situations than the existing schemes, most of which deal with binary images. We believe our technology can lay the foundation for the future.

**Author Contributions:** Conceptualization, J.S.-T.J. and Y.-H.C.; methodology, J.S.-T.J.; software, Y.-H.C.; validation, J.S.-T.J. and Y.-H.C.; formal analysis, Y.-H.C.; investigation, J.S.-T.J.; data curation, Y.-H.C.; writing—original draft preparation, Y.-H.C.; writing—review and editing, J.S.-T.J.; visualization, Y.-H.C.; supervision, J.S.-T.J.; project administration, J.S.-T.J.; funding acquisition, J.S.-T.J. and Y.-H.C. All authors have read and agreed to the published version of the manuscript.

**Funding:** This research was funded by Ministry of Science and Technology of the Republic of China grant number MOST 110-2221-E-260-003, and 111-2813-C-260-033-E. And The APC was funded by MOST 110-2221-E-260-003, MOST 111-2115-M-260-001 and 111-2813-C-260-033-E.

**Institutional Review Board Statement:** Not applicable.

**Informed Consent Statement:** Not applicable.

**Data Availability Statement:** Not applicable.

**Acknowledgments:** We would like to thank the referees for their careful reading of the manuscript and fruitful comments.

**Conflicts of Interest:** The authors declare no conflict of interest.

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
