# Peer review of "XOR-Based (n, n) Visual Cryptography Schemes for Grayscale or Color Images with Meaningful Shares"

_applsci, doi:10.3390/app121910096_

Round 1
Reviewer 1 Report
Dear Authors,
Thank you for your good paper. Some concerns in detail are given as follows:
· There are grammatical errors in the paper. The authors require a native speaker to proofread. The authors can use the professional version of the Grammarly system.
· The abstract is not well written. Please edit the content of the “results” part. Please add the values of the results.
· The authors should update the review to 2022.
· The research gap must be described at the end of the introduction section. Then, the authors explain their work briefly.
· Please add the following sentence at the end of the introduction section:
The rest of the paper is organized as follows: Section two describes …
Author Response
We would like to thank the referees for their careful reading of the manuscript and fruitful comments. We already modify and rewrite our manuscript carefully. The responses to each comment can be found in the attached file.

Reviewer 2 Report
This work proposes and evaluates a XOR-based visual cryptography scheme by which an image can be split into multiple shares hidden behind a cover image. The work is well written but may benefit from some fixes:
1. The graphical quality of Figure 7 has to be improved.
2. The case when k out of the n principals that hold the shares are corrupted has to be discussed. What happens if part of the shareholders collude, can they still recover part of the hidden image?
3. Line 2 of Algorithm 6 says “For each channel A {R, G, B} do Step 2-4.”. Do you mean do Steps 3-4?
4. In Section 3.1 the authors explain the role of the PSNR (higher than 30 to be undetectable by human eye) but no such justification is given for the use of MSE. It should be explained right from the beginning that the MSE covers the distance from the original image and the shares and the expectations for this value have to introduce (as done for PSNR).
5. The comparison in Table 8 reveals that there is no difference in terms of capabilities between the current proposal and related work in [9]. Perhaps the authors could comment more, outlining some advantages that favor their approach.
6. The references list is very short, this may unfortunately suggest that there is not much interest in this area. The authors should expand this list with more references, perhaps adding works that target the applicative area, e.g., images stored on smartphones or social networks, etc.
Otherwise, the paper is interesting to read and well written.
Author Response

(The authors gave the same response as above.)

Round 2
Reviewer 2 Report
The authors responded to my inquiries, I have no further comments.